# A Data Analytics Approach to Assess the Functional and Physical Performance of Female Soccer Players: A Cohort Design

**DOI:** 10.3390/ijerph19158941

**Published:** 2022-07-22

**Authors:** Francisco Tomás González-Fernández, Alfonso Castillo-Rodríguez, Lorena Rodríguez-García, Filipe Manuel Clemente, Ana Filipa Silva

**Affiliations:** 1Department of Physical Education and Sports, Faculty of Education and Sport Sciences, Campus Melilla, University of Granada, 52006 Melilla, Spain; ftgonzalez@gmail.com or; 2Department of Physical Education and Sports, University of Granada, 18010 Granada, Spain; acastillo@ugr.es; 3Department of Physical Activity and Sport Sciences, Pontifical University of Comillas, 07013 Palma, Spain; lrodriguez@cesag.org; 4Escola Superior Desporto e Lazer, Instituto Politécnico de Viana do Castelo, Rua Escola Industrial e Comercial de Nun’Álvares, 4900-347 Viana do Castelo, Portugal; anafilsilva@gmail.com; 5Research Center in Sports Performance, Recreation, Innovation and Technology (SPRINT), 4960-320 Melgaço, Portugal; 6Instituto de Telecomunicações, Delegação da Covilhã, 1049-001 Lisboa, Portugal; 7The Research Centre in Sports Sciences, Health Sciences and Human Development (CIDESD), 5001-801 Vila Real, Portugal

**Keywords:** football, functional performance, physical performance, sports training, female soccer players

## Abstract

Background and Objectives: The aim of this study was twofold: (i) to observe the individual results of fitness status [countermovement jump (CMJ)], hop test, linear sprinting time, stride frequency, stride distance, force–power–velocity, peak power maximal and maximal speed, and repeated sprint ability (RSA) and functional performance (overhead squat, single-leg squat test, dumbbell hip-hinge, Thomas test, hip extension, and internal and external hip rotators) and (ii) to analyze the relationship between anthropometrical measures and values of fitness status with % of difference in the Thomas test, hip extension, and internal and external hip rotators. Materials and Methods: The study followed a cohort design. Sixteen female soccer players competing in the second division of the Spanish league were monitored during the first days of the pre-season. These players were evaluated three times during the preseason of the cohort period. The dataset showed a negative moderate correlation between time and percentage of difference in hip angle and another positive moderate correlation between maximal sprint and percentage of difference in hip angle (r = −0.54, *p* = 0.02 and r = 0.53, *p* = 0.04), respectively. The correlations of stride time and distance with hip extension are interesting, as professional soccer players should have higher levels of hip flexor flexibility during the stride phase, recoil of the instep kick action.

## 1. Introduction

Interest in women’s soccer has grown exponentially in the last few years [1]. Financial support from the Union of European Football Association (UEFA) has trebled [2] and participation rates over the last 10 years have grown by a third [3]. Worldwide, the Fédération International de Football Association (FIFA) is committed to increasing the number of female soccer players from ~13.3 million (2019) to 60 million by 2026 [1]. For this reason, many investigators have taken this into consideration, resulting in a rapid growth of the interest in the links between performance and female soccer players in the last two decades [3]. As is well known, soccer is considered a contact sport and this has had consequences, involving both a greater skill level and physical demands throughout training and matches [4]. Due to its growing popularity, female soccer players are exposed to greater training volumes and competition demands than ever before and, therefore, a better understanding of female players’ physical performance changes is needed to design appropriate training programs [4]. In this context, it is necessary to comment that women’s soccer is being studied with great interest to analyze the changes produced.

In women’s soccer, sprinting is considered a high-intensity effort [5] and high-speed activity is an essential component of matches. Usually, such efforts occur during decisive moments in the match [6], though they represent only 8% to 12% of the total distance covered in a typical match [7]. Additionally, female soccer players were found to perform between 70 and 190 high-intensity runs (>19.8 km·h^−1^) during a match [8,9], covering between 210 and 520 m [6]. In fact, they perform between 1350 and 1650 changes in activity, such as passing, tackling, trapping, and dribbling [8,9]. Accordingly, the capacity to perform maximal sprinting and repeated sprint ability in female soccer players is key for highlighting the individual capacity of each player [10].

Maximum sprint speed and high acceleration are two important components of sprint performance that could determine success in a decisive situation in soccer and enable winning the ball from opponents [11]. These capacities are a topic of great interest to coaches to properly structure and control the training load. Repeated-sprint ability (RSA) is the capacity to repeatedly produce maximal or submaximal sprints spaced over time with short recoveries during a game [12]. Consequently, female soccer players’ ability to withstand repeated maximal sprint efforts is essential to provide better athletic performance [11,13]. A review of the literature reveals that the performance of RSA has a strong correlation with sprinting skills and high-intensity performance in professional soccer players [9]. For this reason, RSA and linear sprint characteristics cannot be separated to investigate which factor is more determinant.

Crucially, to sustain the requirements and demands of a match, female soccer players should show a developed fitness status [14]. A review of the literature reveals that female soccer players must possess high values of maximal oxygen uptake to be able to maintain all high-intensity efforts during the match [7]. In this respect, it should be noted that high-intensity efforts, normally shown in maximal sprints, are performed during a critical moment of the competition [6]. Indeed, the direction and magnitude of high-intensity effort seem to focus on the quantity and quality of sprints during the match [15] and the quantity of meters covered [16]. In addition to sprinting capacity, jumping and hopping power are determinants of soccer performance. In fact, jump power is a greater predictor of sprint ability [17]. Predictions between jump ability and sprinting performance have been investigated and, in this regard, the relationships between hop (horizontal) and jump (vertical), maximizing running speed performance, stride length (SL), stride frequency (SF), and the kinematics of sprinting, have received a great deal of attention in the scientific literature over the years [18]. Therefore, it seems decisive that the factors linked to female soccer players’ performance will be related to the good physical condition and health of the athlete.

Considering the above, assessment of the quality of movement could be essential to improve biomechanical factors that are directly related to economy of movement, motor control, and the effectiveness and prevention of injuries. For this reason, we can highlight that the relationship between performance and functional assessment is crucial to improve the soccer behavior of athletes. In addition, considering the popularity of describing the physical demands of semi-professional and professional soccer players during matches and the increase in organizations, coaches, and practitioners, the present topic is of crucial interest in the scientific literature [19].

Despite the increased participation of women in elite soccer, studies investigating injuries in this cohort are scarce [20]. Soccer is a complex, high-intensity contact sport associated with considerable risk of injury [21]. Epidemiological studies in professional soccer showed an incidence of 10–35 injuries per 1000 h of play, while women recorded 9.1–24 injuries per 1000 h of play. Although the overall injury incidence in women was lower than in men, they recorded a higher number of severe injuries [22].

The predominance of one limb over the other is usual in many athletes, and for that reason, asymmetries between limbs are commonly seen in different sports [23,24]. Actually, in some cases, being asymmetric seems to play an important role for performance, in particular, in elite sports [25]. Therefore, a given level of asymmetry may be considered functional. However, there is a natural threshold in which the asymmetry may lead to an increase in risk for each player [26]. Possibly, this threshold is individual for each player and it is important to consider meaningful variations in this in intra-individual analysis [25].

In order to improve sports performance and injury prevention, the previous evaluation of players represents a crucial aspect of sport practice in general, and soccer in particular. Physical assessment helps sport professionals to identify deficits in athletes that may pose a threat, individualize injury prevention strategies, and set sport–physical performance goals [27]. Many studies have addressed what tests should be performed to detect deficits that threaten athletes’ performance and increase the risk of injury [28]. Therefore, the risk of injury in professional female soccer players is multifactorial, complex, and associated with various intrinsic and extrinsic factors, presenting certain differences with respect to the male sex. Therefore, it is necessary to continue investigating the different risk factors to determine an effective system of evaluation for women athletes to detect their risk of injury [29].

In summary, many researchers have taken this into account, resulting in a rapid growth of interest in the relationships between RSA and maximal sprint in female soccer players. However, to our knowledge, very few studies have investigated the influence of strength–power–velocity, vertical and horizontal jumps, and repeated sprint capacity on the sprint performance of adult female soccer players. Knowing the relationships between jumping/hopping performance and sprinting variables could reveal which indicators are the most relevant for coaches to refine their training prescription and identify which jumping and jumping exercises should be incorporated into a daily routine to improve sprint performance [18]. On the basis of previous research, the purposes of this study design implemented with female soccer players were (i) to observe the results of fitness status (CMJ, hop test, linear sprinting time, stride frequency, stride distance, force–power–velocity, peak power maximal and maximal speed, and RSA) and functional performance (overhead squat, single-leg squat test, dumbbell hip-hinge, Thomas test, hip extension and internal and external hip rotators) and (ii) to analyze the relationship between anthropometrical measures and values of fitness status with % of difference in the Thomas test, hip extension, and internal and external hip rotators. In this regard, the hypothesis of this paper is that better values of anthropometrical measures and values of fitness status in female soccer players may determine less % of difference in the tests of functional performance analyzed.

## 2. Materials and Methods

### 2.1. Study Design

An observational analytic cohort design was used in the present research. Female soccer players were recruited at the end of the 2020–2021 season. Fitness assessments were carried out three times during the preseason. The study was conducted between September and November 2021. In the first session, the functional assessment was made: (i) overhead squat, (ii) single-leg squat test, (iii) dumbbell hip-hinge, (iv) Thomas test, and (v) hip extension and internal and external hip rotators. In the second and third sessions, fitness status was assessed. CMJ, hop test, linear sprinting time, stride frequency, stride distance, force–power–velocity, peak power maximal and maximal speed were evaluated on the second and third days that the RSA test was performed. All the participants completed all the sessions of assessment.

Regarding the day of training, participants trained twice a week (90 min per session) and played one official match a week. Generally, training sessions comprised a warm-up, main part, and cool down. Crucially, at the moment of assessment the team had no official matches. Familiarization with each test was given during two weeks of the preseason.

All the participants in this research were informed about the main aims of the study and completed and signed informed consent forms prior to the first session. The athletes were treated according to American Psychological Association (APA) guidelines, which ensured the anonymity of participants’ responses. In addition, the study was conducted in accordance with the ethical principles of the Helsinki Declaration for Human Research and was approved by a scientific council of the local university (code: 2021/85).

### 2.2. Participants

In total, sixteen female soccer players from one team in the second division of the Spanish league participated in the present study (age: 21.00 ± 4.32 years; mass: 60.07 ± 7.07 kg; height: 164.63 ± 5.71 cm) from the Baleares region, Spain, recruited from the city of Palma de Mallorca, which has a population in the range of 250,000 to 500,000 inhabitants according to the Spanish Government National Institute of Statistics (http://www.ine.es/; accessed on 29 June 2022). Concerning the sample size, the following equation was used: Sample Size = *Z*2 × (*p*) × (1 − *p*)/*C*2, where *Z* = confidence level (95%); *p* = 0.05 and *C* = margin of error 0.05. See Table 1 for more information.

The inclusion criteria for this study were: (i) a background of ≥5 years of systematic soccer training and competitive experience, (ii) continuous soccer training for the previous 3 months and players who had had injuries or illness no longer than 4 consecutive weeks, and (iii) absence of potential medical problems. In the case that any participants did not comply with one or more inclusion criteria they were removed from the study. Lastly, these players trained twice a week (90 min per session) and played one official match a week. Generally, training sessions comprised a warm-up, main part, and cool down.

### 2.3. Procedure

#### Data Collection

The functional and fitness assessments were recorded in three different sessions before the start of the training session (7:30 to 9:30 p.m.). It is relevant here that assessments were performed in the preseason and were divided into three sessions: (i) anthropometric measurements and all tests of functional assessment, (ii) overhead squat, (iii) single-leg squat test, (iv) dumbbell hip-hinge, (v) Thomas test, and (vi) hip extension and internal and external hip rotators. This session was performed in a private room near the locker room with a stable temperature of 22 °C and relative humidity of 52%. In the second and third sessions, fitness status was assessed (second session (CMJ, hop test, linear sprinting time, stride frequency, stride distance, force–power–velocity, peak power maximal and maximal speed) and in the third session the RSA test). Fitness assessments were performed on a synthetic turf field with a mean temperature of 23.5 °C and relative humidity of 70 °C ± 3% and no windy or rainy conditions were experienced during the assessments.

Before each assessment, the participants performed a warm-up consisting of different phases: a general activation phase with five minutes of jogging, general and specific joint mobility, dynamic stretching, and a gesture-specific warm-up. The warm-up consisted of general joint mobility, five minutes of running at a moderate speed, and the following self-loading exercises: 10 squats including heel raises at the end of the gesture, 10 pelvic anteversion–retroversion movements, 10 hip external–internal rotation movements in standing position, and 10 monopodal deadlifts.

The exercises were recorded for qualitative analysis with the HD camera of an iPad Pro model A1673 (iOS 13.3) attached to a tripod, and subsequently the videos were analyzed with Kinovea Software version 0.8.15 (Joan Charmant; Bordeaux, France) [30] in an LCD 15.6″ laptop. The deep squat and dumbbell hip-hinge tests were performed using a plastic stick. The Thomas test, hip extension, and rotator tests were performed on a medical stretcher. For the comparison of both legs, the dominant limb (DL), as opposed to the non-dominant (NDL), was established as the preferred leg to kick the ball (see Figure 1 for more information).

### 2.4. Outcome Measures

#### 2.4.1. Anthropometry

Mass (kg) was measured without shoes with a bioelectrical impedance analysis (BIA) device (Tanita BC-730) to the nearest 0.1 kg. Height (cm) was measured using a stadiometer (Type SECA 225, Hamburg, Germany) to the nearest 0.1 cm. Body mass index (kg/m^2^) was calculated as mass (in kilograms) divided by height squared (in meters).

#### 2.4.2. Countermovement Jump

The CMJ was evaluated using the jump platform Chronojump-Boscosystem^®^ (Barcelona, Spain) developed by de Blas et al. (2012) [31], which revealed intraclass correlation test levels between 0.821 and 0.949 to measure jump height. This system was connected to a MacBook Pro (macOS South 11.1). Furthermore, the measurements were analyzed by Chronopic and recorded by Chronojump version 2.0.2. The female soccer players performed three jumps with 30 s of recovery between attempts to minimize the effect of fatigue. They were instructed to jump as high as possible after reaching a knee angle of ~90°. In addition, participants were also instructed to keep their hands on their hips during the CMJ and to land with their legs extended with maximum foot plantar flexion. If any of these requirements were not met, the jump was repeated.

#### 2.4.3. Hop Test

Before the start of the hop test, two practice jumps were performed to familiarize the subjects with the tests. Subsequently, these tests were followed by three official tests with dominant leg and the non-dominant leg, with 3 min of rest between them, for data collection. To perform the jumps, all the participants were instructed to perform the hop test dominant and hop test non-dominant with arm swing. For both hop tests, each participant jumped with one leg, attempting to jump as far as possible with a single jump. The best jump was retained for further analysis. For more information, see the manuscript of Rosch et al., 2000 about assessment and evaluation of football performance [32].

#### 2.4.4. Linear Sprint

The aim of the present test was to run 30 m as fast as possible. The 30 m sprints were evaluated employing the *MySprint* app. To ensure a successful performance, we followed the protocol of Samozino et al., 2016 [33]. Thus, participants sprinted at maximum speed and were given two attempts. We recorded the best of the two attempts (measured in seconds by the MySprint app and iPad Pro model A1673 (iOS 13.3). A camera (HD 1080 p 240 fps) was used to record and analyze all attempts.

#### 2.4.5. Repeated-Sprint Ability Test by Bangsbo with a Change-of-Direction Test

The protocol used for testing RSA consisted of sprinting 30 linear meters (with change of direction), performed 7 times and with a recovery time between efforts of 10 s. We followed the protocol established by Bangsbo 1994 [29] but with a modified version with change of direction. Microgate Wireless Training Timers (Microgate, Bolzano—Italy), with digital FSK transmission, redundant code with information correctness verification and autocorrection, multifrequency transceiver 433–434 MHz and impulse transmission accuracy ±0.4 ms, were positioned at the beginning and end lines to record the time of each sprint. The participants started their sprint 0.5 m behind the start timing gate. The time (s) for each trial was recorded. After that, minimum and maximum peak power were determined using the equation [34] $\mathrm{Power}\;=\frac{\mathrm{Body}\;\mathrm{mass}\;\times \;{\mathrm{Distance}}^{2}}{{\mathrm{Time}}^{3}}$, and the fatigue index used the following equation $\mathrm{Fatigue}\;\mathrm{index}=\frac{\max_{\mathrm{power}}\;-\;\min_{\mathrm{power}}\;}{\mathrm{Sum}\;\mathrm{of}\;6\;\mathrm{sprints}\;\left(s\right)}$.

### 2.5. Functional Assessment

#### 2.5.1. Overhead Squat (DS)

The female soccer players were assessed using the DS test, outlined in Functional Movement Systems (FMS™) [35,36]. Faulty movement patterns and compensations can then be identified more readily, using the FMS scale taking into account that trials were scored on a rising scale from 0 to 3. A 0 score indicated that pain was reported during the exercise. In the case of a 1 score it revealed failure to complete the exercise or loss of balance. A 2 score indicated that the exercise was completed with heel raise compensation. Lastly, a 3 score showed the performance of the movement without compensation [37].

#### 2.5.2. Single-Leg Squat Test (SLS)

For SLS, the female soccer players were asked to stand on one leg facing the camera and were instructed to squat down as low as possible. Accordingly, the knee was flexed between 60° and 80° and this position was retained for five seconds. Consecutively, participant had to return to the initial position. A score was assigned to the performance of the test from 1 to 10 points. To record and assess data, we followed the protocol established by Herrington et al., 2013 [38].

#### 2.5.3. Dumbbell Hip-Hinge (HHD)

The female soccer players were asked to stand upright while holding a dumbbell behind their back in contact with the sacrum, between the scapulae and behind the head. They performed a deep hip flexion with minimal knee flexion while holding the dumbbell in contact as explained. Two repetitions were completed for each participant. The ability to achieve a 90° movement of the dumbbell relative to the femur in the sagittal plane while maintaining lumbar curvature and the dumbbell with all 3 contacts was observed (Liebenson, 2003) [39].

#### 2.5.4. Thomas Test (TT)

The Thomas test (TT) is a pass/fail test in which the patient lies supine upon an examination table with both legs straight out in front of them on the tabletop. While supine, the patient flexes the hip of one leg and holds the knee of the same leg maximally flexed at the chest. The pelvis is maintained in neutral throughout. The contralateral leg is allowed to remain relaxed and flat against the tabletop. A positive TT, which is taken as indicative of hip flexion contracture, is where there is noticeable hip flexion of the contralateral leg, as indicated by a gap between this leg and the tabletop.

The female soccer players lay supine on a table so that their gluteal fold was located at the end of the table and they held both knees to their chest. Then they lowered their right leg until it came to rest, at which point they were instructed to keep the leg as relaxed as possible. Thomas-test joint angles were calculated from a leg position corresponding with a posterior pelvic tilt (lumbar spine on table), which was defined as a horizontal alignment of the iliac crest and trochanterion landmarks. If the alignment was judged as not being horizontal, the subject was asked to move the contralateral knee closer to or farther from the chest. To record the data, we followed the protocol established by Magee (2013) [40].

#### 2.5.5. Hip Extension (with Knee Extension and Knee Flexion) (HEKE and HEKF)

For HEKE and HEKF, the female soccer players were asked to lie in a prone position, with both arms relaxed and hands resting under the forehead. We followed the protocol established by Magee (2013) [40]. No compensatory movements such as lumbar extension or pushing the stretcher with the arms were allowed during the action. The researcher controlled the pelvis posteriorly to stabilize it during the test. The test was performed with the knee extended and with the knee flexed for each leg. Two repetitions of both tests were completed for each limb.

#### 2.5.6. Internal and External Hip Rotators (IHR and EHR)

The subjects were asked to lie in a prone position with both arms relaxed and hands resting under the forehead. The researcher controlled the posterosuperior iliac crest of the pelvis to stabilize it during the test. The landmarks used to measure the angle were the medial point of the ankle and patella. Two tests were completed for each leg and each movement. We followed the protocol established by Magee (2013) [40].

#### 2.5.7. Statistical Procedures

Descriptive statistics were calculated for each variable (see Table 1 for more information). Before any parametric statistical analysis was performed, the assumption of normality was tested with the Kolmogorov–Smirnov test on each variable. A Pearson correlation coefficient r was used to examine the relationship between anthropometrical measures (age, mass, height, body mass index, lean mass, and muscle), countermovement jump, hop test (dominant and non-dominant), linear sprinting 30 m (time, stride frequency and stride length), force–power–velocity (Pmax and Maximal speed), RSA test (Pmin, Pmax, FI) and Thomas test (percentage of difference in hip angle and percentage of difference in knee angle), hip extension (percentage of difference in hip extension (knee flexion) and percentage of difference in hip extension (knee extension)) and internal and external hip rotators (percentage of difference in internal rotators), and percentage of difference of external rotators. The interpretation of the d regardless of the sign, followed the scale: very small (0.01), small (0.20), medium (0.50), large (0.80), very large (1.20), and huge (2.0), as initially suggested by Cohen [41] and expanded by Sawilowsky [42]. The level of significance was set at α = 0.05. The results were categorized as poor, regular, and good, taking into account the classifications of each of the physical tests carried out, favoring a quick and efficient interpretation. Data were analyzed using Statistica software (version 13.3; Statsoft, Inc., Tulsa, OK, USA).

## 3. Results

Sixteen female soccer players, from one team from the second division of the Spanish league, were assessed. Descriptive statistics were calculated for each variable. See Table 1 for more information.

### 3.1. Overhead Squat

Overhead squat results showed eight female soccer players (50%) with an excellent score, six (37.5%) with a fair score, and two (12.5%) with a poor score.

### 3.2. Single-Leg Squat Test

Single-leg squat test results showed nine female soccer players (56.25%) with an excellent score, one (6.25%) with a fair score, and six (37.5%) with a poor score.

### 3.3. Dumbbell Hip-Hinge

Dumbbell hip-hinge results showed thirteen (81.25%) female soccer players with an excellent score and three (18.75%) with a fair score.

### 3.4. Thomas Test

The Thomas test results showed, in right hip angle, a total of twelve female soccer players (75.00%) with an excellent score and four (25.00%) with a fair score. In relation to left hip angle, a total of ten female soccer players (62.5%) had an excellent score and six (37.5%) a fair score. In reference to percentage of difference in hip angle, it is necessary to mention that four participants (25%) had a high percentage of difference (>10%), the rest (75%) had a normal percentage (<10%). The percentage of difference in the knee showed eight subjects (50%) had a high percentage of difference (>10%). In fact, three had asymmetries in both segments. Continuing with the percentage of difference in the knee, the data revealed that eight of the players (50%) had less than 10% difference. See Table 2 for more information.

### 3.5. Hip Extension (with Knee Extension and Knee Flexion)

Hip extension results showed a percentage of difference in values with knee flexion as follows: five female soccer players (31.25%) had a high percentage of difference (>10%) and the rest (68.75%) had a moderate percentage (<10%). The percentage of difference with knee extension revealed two subjects (12.5%) had a high percentage of difference (>10%) and only one had asymmetries in both segments. The rest (87.5%) had less than 10% of difference (see Table 2).

### 3.6. Internal and External Hip Rotators

On the one hand, internal hip rotator results showed a high percentage of difference in three female soccer players (18.75%) (>10%); the rest (81.25%) had a moderate percentage (<10%). On the other hand, external hip rotator results revealed a high percentage of difference (>10%) in three subjects (18.75%); the other thirteen (81.25%) had a moderate percentage of difference (<10%). Two subjects showed asymmetries in both assessments (see Table 2).

Subsequently, correlation analyses were performed between anthropometrical measures and % of difference in the Thomas Test (percentage of difference in hip angle and percentage of difference in knee angle), hip extension (percentage of difference in hip extension (knee flexion) and percentage of difference in hip extension (knee extension)), and internal and external hip rotators (percentage of difference in internal rotators and percentage of difference in external rotators). In this respect, no correlations were found.

Other correlation analyses were performed between the countermovement jump, hop test dominant and hop test non-dominant, % of difference in the Thomas test (percentage of difference in hip angle and percentage of difference in knee angle), hip extension (percentage of difference in hip extension (knee flexion) and percentage of difference in hip extension (knee extension)), and internal and external hip rotators (percentage of difference in internal rotators and percentage of difference in external rotators). As with the prior analysis, no correlations were found.

New correlation analyses were performed between linear sprinting (time; stride frequency and stride length) and force–power–velocity values (maximal sprint and Pmax) and % of difference in the Thomas test (percentage of difference in hip angle and percentage of difference in knee angle), hip extension (percentage of difference in hip extension (knee flexion) and percentage of difference in hip extension (knee extension)), and internal and external hip rotators (percentage of difference in internal rotators and percentage of difference in external rotators). The dataset showed a negative moderate correlation between time and percentage of difference in hip angle and another positive moderate correlation between maximal sprint and percentage of difference in hip angle (r = −0.54, *p* = 0.02 and r = 0.53, *p* = 0.04), respectively (see Table 3).

Finally, more correlation analyses were performed between RSA (Pmin, Pmax and FI) and % of difference in the Thomas test (percentage of difference in hip angle and percentage of difference in knee angle), hip extension (percentage of difference in hip extension (knee flexion) and percentage of difference in hip extension (knee extension)), and internal and external hip rotators (percentage of difference in internal rotators and percentage of difference in external rotators). No correlations were found.

## 4. Discussion

The aim of this study was to assess fitness status and analyze the relationship between anthropometric measurements and physical status with percentage of difference in the Thomas test, hip extension, and internal and external hip rotators in female soccer players. The results made it possible to report on the body characteristics and physical capacity of different female players. More than 50% of the players showed a good performance in the functional performance tests: overhead squat, single-leg squat test, dumbbell hip-hinge, and Thomas test. Regarding the second objective, there were no correlations between anthropometrical measures, jump test and RSA performance, and functional performance tests. However, there was a negative moderate correlation between time and percentage of difference in hip angle and another positive moderate correlation between maximal sprint and percentage of difference in hip angle. Taking these results into account, the main finding of the study is that there is no clear evidence of relationships between the functional performance score and physical condition, except for those discussed above, which is consistent with the study by Parchmann and McBride (2011) [43].

Regarding the description of the evaluation of physical condition, they obtained a much lower CMJ performance than American university soccer players with similar ages (between 18 and 23 years [44,45], although they are similar to other Spanish U17 soccer players [46]. Regarding the performance in the linear sprint test at 30 m, the result is similar to Norwegian U17 female soccer players [47] and better than soccer players in the Italian national team, whose average age is 25 years [48]. This difference in physical condition in soccer players of the same skill level or category could possibly be due to the difference in investment between the different countries in the sport of soccer [49].

The evaluation of functional performance was confirmed to be good in the majority of the female soccer players evaluated, being scientifically associated with the quality of the movement patterns [50]. However, deficits in each of the functional performance tests were observed in a small number of soccer players [51], confirming heterogeneity among these athletes. In addition to this, some asymmetries were verified in these functional performance tests, sometimes exceeding 7%. These asymmetries have been associated with an increased risk of injury from 8% on measurements of the lower limbs [52]. In the same way, the threshold of 10% is considered for asymmetries in physical condition that are associated with the risk of injury [53]. These asymmetries could explain a decrease in physical performance [54] and, therefore, we present it as a possible limitation of the study.

The moderately powered (r > 0.50) correlations of stride time and distance with hip extension are interesting, as professional soccer players should have higher levels of hip flexor flexibility during the stride phase, recoil of the instep kick action [55]. Taking into account the initial hypothesis of this study, which indicated that the best values of the anthropometric measurements and values of the physical state could determine a lower percentage of difference in the functional performance analyzed, there were no conclusive findings, thus, rejecting this hypothesis.

There are several strengths to the present study. Firstly, it described the physical condition of a semi-professional women’s football team in Spain. In addition, it was possible to verify the degree of functional and physical performance, in addition to the corporal characteristics that they present. Various asymmetries were described that could condition sports performance. This study provides the added value that the entire physical and physiological profile has been brought together in this research.

Several limitations exist in this study. It has already been commented in this section that a description of the asymmetries in the soccer players has been given, which could cause a decrease in performance. Future research is urged to increase the sample number in different sports clubs and to study the effect of asymmetries on sports performance, both in competition and in physical tests in training sessions. Another limitation could be the consideration of the biological age of the athlete to make the correlations, since the stages of maturation could be determined instead. Another important limitation was the potential bias or imprecision, although the material and protocol used were fully respected and only one researcher was responsible for the data recording. Data obtained should be interpreted with caution and the general dimension of the sample considered. Finally, due to the direct objectives with functional and physical performance profiles, it would be desirable to study the fluctuations at different times of the season, as some players may be found with a high degree of fatigue at times corresponding to the beginning of the season (after a month and a half preseason with a high level of physical work).

## 5. Conclusions

In conclusion, this study described the physical and functional performance of semi-professional female soccer players, evaluating the correlations between both areas. The main finding of this study is the description of high asymmetries in some tests of physical condition and functional performance. In addition, some correlations were found between these functional performance variables and the anthropometric and physical characteristics of the athletes. This paper provides knowledge about physical condition and functional performance in female soccer players to improve individual monitoring to provide the most adequate strategies for technical and physical staff. The comprehensive model could be strengthened by increasing knowledge about injury risk. As a practical application, the inclusion, in individual or subgroup training programs, of tasks related to hip extension is recommended, due to its direct relationship with physical responses related to sprinting.

## Figures and Tables

**Figure 1 ijerph-19-08941-f001:**
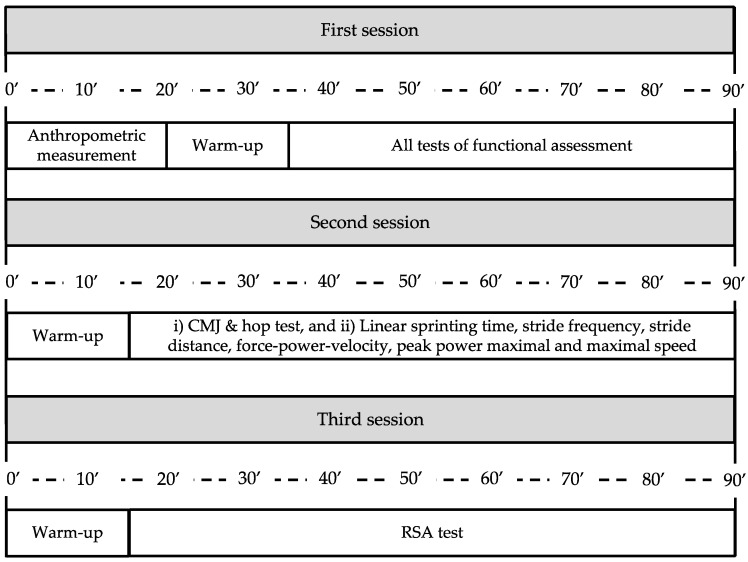
Schematic representation of study (see text for full description).

**Table 1 ijerph-19-08941-t001:** Anthropometric measurements and fitness variables at the assessment (mean ± SD).

Female Soccer Players (*n* = 16)				
	Assessment	LCI 95%	CI 95%	UCI 95%
Anthropometric measurements				
Age (yrs)	21.00 ± 4.32	18.88	2.12	23.12
Mass (kg)	60.07 ± 7.01	56.64	3.43	63.50
Height (cm)	164.63 ± 5.71	161.83	2.80	167.43
Body Mass Index (kg/m^2^)	22.11 ± 2.00	21.13	0.98	23.09
Lean Mass (%)	28.82 ± 3.98	26.87	1.95	30.77
Muscle (%)	30.56 ± 1.94	29.61	0.95	31.51
Countermovement jump				
CMJ (cm)	23.80 ± 2.27	22.68	1.11	24.91
Hop Test				
Dominant (cm)	121.92 ± 7.57	118.21	3.71	125.63
Non-Dominant (cm)	123.89 ± 9.28	119.34	4.55	128.44
Linear Sprinting (30 m)				
Time (s)	5.20 ± 0.18	5.12	0.09	5.29
Stride frequency (*n*)	3.76 ± 0.21	3.65	0.10	3.86
Stride length (m)	1.54 ± 0.09	1.49	0.04	1.58
Force–power–velocity				
PMax (W/kg)	13.43 ± 1.88	12.51	0.92	14.35
Maximal Speed	6.97 ± 0.31	6.82	0.15	7.12
RSA test				
Pmin (s)	138.16 ± 22.06	127.35	10.81	148.97
Pmax (s)	165.72 ± 22.16	154.86	10.86	176.58
FI (%)	0.55 ± 0.19	0.46	0.64	0.09

Note: UCI: Upper confidence interval; CI: Confidence interval; LCI: Lower confidence interval; CMJ: Countermovement jump; 30 m: 30-m sprint; Force–power–velocity: Peak power (Pmax) and maximal sprint; RSA: Repeated-sprint ability; Pmin: Peak power (minimum); Pmax: Peak power (maximum); FI: Fatigue index.

**Table 2 ijerph-19-08941-t002:** Thomas test, hip extension (with knee extension and knee flexion), and internal and external hip rotators assessment.

Thomas Test							
Frontal View		Frontal View					
RA	LA	Right hip angle	Left hip angle	% of difference	Right knee angle	Left knee angle	% of difference
Mean ± SD		92.19 ± 6.29	102.13 ± 7.71	4.44 ± 2.60	96.28 ± 4.86	105.25 ± 7.54	4.20 ± 2.27
**Hip Extension (with Knee Extension and Knee Flexion)**							
		**Internal Rotation**			**External Rotation**		
		Hip extension (right knee flexion at 90°)	Hip extension (left knee flexion at 90°)	% of difference	Hip extension (right knee extension)	Hip extension (left knee extension)	% of difference
Mean ± SD		28.22 ± 6.74	32.97 ± 8.19	2.87 ± 1.74	36.13 ± 6.02	38.00 ± 9.66	2.12 ± 2.25
**Internal and External Hip Rotators**							
		**Internal Rotation**			**External Rotation**		
		Right internal rotation	Left internal rotation	% of difference	Right external rotation	Left external rotation	% of difference
Mean ± SD		38.09 ± 7.00	36.19 ± 9.89	2.56 ± 2.18	42.75 ± 7.02	43.28 ± 8.29	2.20 ± 2.28

**Table 3 ijerph-19-08941-t003:** Pearson’s correlation coefficient between linear sprinting and force–power–velocity values and % of difference in the Thomas test, hip extension, and internal and external hip rotators (*n* = 16).

Female Soccer Players (*n* = 16)						
	Thomas Test		Hip Ext.		Int. and Ext. Hip Rotators	
	% Dif of Hip Angle	% Dif of Knee Angle	% Dif of Hip Ext. (Knee Flexion)	% Dif of Hip Ext. (Knee Ext.)	% Dif of Int. Rotation	% Dif of Ext. Rotation
**Time (s)**	**r = −0.57**	r = 0.03	r = −0.01	r = −0.30	r = 0.34	r = −0.02
	** *p* ** **= 0.** **02 ***	*p* = 0.90	*p* = 0.98	*p* = 0.26	*p* = 0.20	*p* = 0.94
**Stride distance**	r = 0.32	r = 0.07	r = −0.2721	r = 0.13	r = 0.18	r = 0.08
	*p* = 0.23	*p* = 0.80	*p* = 0.32	*p* = 0.64	*p* = 0.50	*p* = 0.75
**Stride frequency**	r = 0.06	r = −0.01	r = 0.28	r = 0.02	r = −0.44	r = −0.05
	*p* = 0.82	*p* = 0.94	*p* = 0.30	*p* = 0.93	*p* = 0.10	*p* = 0.85
**Max sprint**	**r = 0.53**	r = 0.12	r = −0.22	r = −0.01	r = −0.33	r = 0.12
	***p* = 0.04 ***	*p* = 0.65	*p* = 0.42	*p* = 0.99	*p* = 0.22	*p* = 0.65
**Pmax (w/kg)**	r = 0.13	r = −0.17	r = 0.17	r = 0.34	r = −0.31	r = −0.47
	*p* = 0.62	*p* = 0.54	*p* = 0.52	*p* = 0.21	*p* = 0.25	*p* = 0.07

Note: Max Sprint: Maximal sprint; % Dif: Percentage of difference; Ext: Extension; Int: Internal; Ext: External * Denotes significance at *p* < 0.05.

## Data Availability

The datasets generated and analyzed during the current study are available from F.T.G.-F. on reasonable request.

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
