# Peer review of "A Data Analytics Approach to Assess the Functional and Physical Performance of Female Soccer Players: A Cohort Design"

_ijerph, 2022, doi:10.3390/ijerph19158941_

Round 1

Reviewer 1 Report

The current study does not provide new or interesting information. It is just a descriptive study of a WSP team in the second division of Spain that does not provide any extrapolated information about the WSP population. Authors try to enhance the quality of the article by presenting individual results to show more data and also interpret values of r=0.53 as large correlations. I think that this kind of descriptive studies are not suitable for publication in this journal due to its lack of scientific soundness. As well, the presentation of the paper has low quality due to the presence of many format mistakes mostly in the citation style.

Line 21. The authors are indicating the abbreviation CMJ and it has not been detailed before in the text. Detail the abbreviation. Same with RSA. These appear again in lines 29 and 30.

In line 30 there is a bracket after RSA that must be erased, as well there is a bracket that must be erased in line 31

This part of the abstract is already mentioned in the objectives, you can erase it "The measured parameters included CMJ, hop test, linear sprinting time, stride frequency, stride distance, force-power-velocity, peak power maximal and maximal speed, and RSA) and functional performance (overhead Squat, Single-leg squat test, Hip-hinge with dumble test, Thomas test, hip extension, and internal and external hip rotators jump"-

Line 34. Are authors considering r=0.54 a large correlation? an r-value of 0.5 must be interpreted as a medium-low correlation in a bivariate correlation. Define in the methods section of the abstract the correlation coefficient used in the study.

In the abstract, the authors draw this conclusion: "The main finding of this study is the description of high asymmetries in some tests of physical condition and functional performance", however, in the results section, they only talk about the correlations.

Line 47. There is a bracket with no sense [

Some citations are done with these brackets (), others with [] and others with [(). This is not the format asked by the journal.

Line 59. it is used a different font.

Line 74. Indicate if this correlation was found in women or men.

Line 173. CMJ has not been described

Line 218. Indicate units for BMI.

Line 319. Correlations can't be interpreted as large with r-values below 0.7 or 0.8.  r<0,3 should be trivial, and 0.3<r<0.5 low. Maybe moderate as 0.6<r<0.7. But never large with values below 0.7.

In the results section, it does not make sense to present Table 1 and figure as they present the same data. Leave only one of them. I think it would be better to leave table 1 as the individual values are not significant in the research.

2.5.1 What was the weight used in the overhead squat?

In the result section, tables 2, 3, and 4 do not provide useful information as the sample is not consistent. These data can't be extrapolated to the general population of WSP.

In general, the results section is not consistent since the authors are providing individual values for each variable. It should be presented only the mean values as individual values do not give any valuable information.

Author Response

REVIEWER 1

  • The current study does not provide new or interesting information. It is just a descriptive study of a WSP team in the second division of Spain that does not provide any extrapolated information about the WSP population. Authors try to enhance the quality of the article by presenting individual results to show more data and also interpret values of r=0.53 as large correlations. I think that this kind of descriptive studies are not suitable for publication in this journal due to its lack of scientific soundness. As well, the presentation of the paper has low quality due to the presence of many format mistakes mostly in the citation style
  • Dear editor, your feedbacks are highly appreciated. In fact, we believe vehemently that the present research, in his actual format and with all changes performed, is a significant contribution to the existing literature due to the relationship between functional and physical performance in women soccer players.

  • Line 21. The authors are indicating the abbreviation CMJ and it has not been detailed before in the text. Detail the abbreviation. Same with RSA. These appear again in lines 29 and 30.
  • The text was changed accordingly.

  • In line 30 there is a bracket after RSA that must be erased, as well there is a bracket that must be erased in line 31
  • Thank you very much. The different problem with the bracket were updated accordingly.

  • This part of the abstract is already mentioned in the objectives, you can erase it "The measured parameters included CMJ, hop test, linear sprinting time, stride frequency, stride distance, force-power-velocity, peak power maximal and maximal speed, and RSA) and functional performance (overhead Squat, Single-leg squat test, Hip-hinge with dumble test, Thomas test, hip extension, and internal and external hip rotators jump"-
  • Done, the text duplicated was deleted and the abstract improve in his comprehension.

  • Line 34. Are authors considering r=0.54 a large correlation? an r-value of 0.5 must be interpreted as a medium-low correlation in a bivariate correlation. Define in the methods section of the abstract the correlation coefficient used in the study.
  • Thank you for your comment. The information was introduced correctly, and we add the cite Cohen J. 2nd ed. Hillsdale: Lawrence Erlbaum Associates; 1988. Statistical power analysis for the behavioral sciences. [Google Scholar]

  • In the abstract, the authors draw this conclusion: "The main finding of this study is the description of high asymmetries in some tests of physical condition and functional performance", however, in the results section, they only talk about the correlations.
  • Thank you. The abstract section was changed accordingly

  • Line 47. There is a bracket with no sense [
  • Thank you very much. The different problem with the bracket accordingly and references were changed accordingly.

  • Some citations are done with these brackets (), others with [] and others with [(). This is not the format asked by the journal.
  • The different problem with the bracket accordingly and references were changed accordingly.

  • Line 59. it is used a different font.
  • Thank you. The text was changed accordingly.

  • Line 74. Indicate if this correlation was found in women or men.
  • The text was updated.

  • Line 173. CMJ has not been described.
  • The text was updated.

  • Line 218. Indicate units for BMI.
  • Thank, the text was updated with the correct units.

  • Line 319. Correlations can't be interpreted as large with r-values below 0.7 or 0.8.  r<0,3 should be trivial, and 0.3<r<0.5 low. Maybe moderate as 0.6<r<0.7. But never large with values below 0.7.
  • Thank you, we follow the values that above mentioned (Cohen and Bland)

  • In the results section, it does not make sense to present Table 1 and figure as they present the same data. Leave only one of them. I think it would be better to leave table 1 as the individual values are not significant in the research.
  • Thank you very much. I´m agree with your suggestion and the table I was deleted.

  • 5.1 What was the weight used in the overhead squat?
  • We use a peak to perform the exercise. In fact, in all moments we follow the protocol of Cook et al., 2014

  • In the result section, tables 2, 3, and 4 do not provide useful information as the sample is not consistent. These data can't be extrapolated to the general population of WSP.
  • I´m totally agree. However, personally I think that these values could provide information about the female soccer players of second division and a valid way to assess the functional performance of the group.

  • In general, the results section is not consistent since the authors are providing individual values for each variable. It should be presented only the mean values as individual values do not give any valuable information.
  • The results section was updated considering your recommendations.

Reviewer 2 Report

Abstract:

The parameters measured for determining fitness are listed twice in the abstract, I suggest that they only need listing once. 

Some words are capitalised when they are not proper nouns. This is true throughout the paper, particularly when the tests are being listed.

The consequence and meaning of the correlations is not interpreted in the abstract. A statement of what the results mean would be helpful as part of the conclusion of the abstract. 

Introduction:

There is a mix of the word 'female' (sex) and 'woman/women' (gender). It may be helpful to clarify and ensure that the correct term is used.

The sentence on lines 93-98 is very long and it is difficult to interpret what is being communicated. Please can this be reworked. Words such as 'it is necessary to mention' are usually superfluous and can be omitted without any lack of meaning from a sentence.

There seems to be a strong thread about the connection to injury and asymmetries that could lead to injury, and yet injury incidence or history is not collected.

Line 121 - I think this should read 'male sex' and not 'male gender' Sex is biological, gender is expression of identify.

Methods

Line 144 - what is meant by 'al analytic'?

Line 148 - what is 'sene'?

Line 150 - Should it read 'day' where it says 'daty'?

Line 151 – This should read 'All participants completed all sessions and assessments'. The word season is confused for session in several places in the paper.

Line 145 - the correct English would be 'were performed three times during the pre-season'.  and 'In the first season' should read 'In the first session'.

Line 153 - two spelling/typographical errors.

Line 154 - The English in this sentence needs correcting, I'm not entirely clear what is meant here. I think it may mean 'During the period when assessments were undertaken, the participants were not playing any official matches.'

Line 155 - again the English translation is awkward and needs reworking. I think the keywords here are about technique (not 'correct grip').

Line 159-161 - spelling/typographical errors

Line 169 - spelling/typographical errors

Line 171 - spelling/typographical errors

Line 173 - 'an' should be 'and'

Line 176 - 'seasons' should be sessions, this is true further on in the paper and needs attending to. To save repeating this comment, please check all appearances of the word 'season' in the paper and check that it is correct. Many instances should be 'session'.

Line 181 - 'near of looker or with' needs correcting

Line 183 - the fitness status tests do not need listing again.

Throughout 2.3.1 - please clarify the type of temperature measurement (i.e., dry bulb) and the correct units for humidity need to be used (Line 187 'C' is used)

My view of methods is that it should be as clear as possible and not require significant reference to other papers. The citations in lines 181 and 191 do make it quite difficult to understand what is being carried out. With the large number of tests being used in this paper I would recommend that a figure/table/diagram is used to illustrate more clearly what is being measured and when.  

Line 206 - 'Ipad' should read 'iPad'

The procedures are described (line 174-212) and then much is repeated when the measures are described (line 213-313). The repeat of information between the two sections could be improved for further conciseness and better structure.

Line 232 – ‘Hop D’ and ‘Hop non-D’ need explaining and/or referencing.

Line 236 – “…previous suggestions by Rosch…” is too vague.

Line 240 – The MySprint app needs full details.

Line 243 - 'Ipad' should read 'iPad'

Line 244 – “To ensure successful performance, we followed the protocol of…” again, is too vague.

Line 261 – This is the first time ‘FMS’ is mentioned and therefore the full words need to be given alongside the abbreviation.

Line 266 – ‘…heel on the head.’? I’m not sure what this means.

Line 277 - the word 'participant' would be preferable to 'subject'

The Thomas Test is not clearly explained.

Line 311 – full details of the Kinovea Software is needed.

Line 315 - "The mean and standard deviation were used for data processing" it's unclear what this means and how the data was used. More detail is needed for clarity.

Table 1 - age should be presented to the same number of decimal places to that which it was originally collected - this is likely to be 0 decimal places. i.e., 21 ± 4. The same is true for the other measures, were they all recorded to 2 decimal places?

Line 341 - 'In this sense' needs amending, the phrasing is unclear.

Table 2, 3, 4, 5, 6, 7 – I’m not sure what the tables are adding to the narrative of the paper. The inclusion of the individual data, without connecting to other data doesn’t help the reader understand the relationship between the fitness status and functional results. Reporting individual observational data doesn’t offer any insights, analysis or clear interpretation for discussion. Table 8 is a more useful table as it draws together the connections between the results. I would prefer to see the mean, SD, and ranges for the results, presented in a table – these would be useful observations and would add to the data about female soccer.

Discussion

Line 440-444 – It’s not necessary to repeat all the test names again.

Line 466-476 – the paragraph is written in a vague way that makes it difficult to understand and therefore it lacks a clarity of message. I’m not sure if the results of the study are being discussed in relation to the published research, or the solely the published research.

Line 485 – this is the first time the women have been described as ‘semi-professional’, this should be mentioned earlier in the paper.

Line 498 – “the entire physical and physiological profile has been brought together in this research” – I don’t think this profile is well presented in the results. The data is available but not presented in such a way to give a clear profile.

Line 501 – a conclusion should be the profiling of these athletes, but only if the results are presented to give this more clearly.

Author Response

REVIEWER 2

Abstract:

  • The parameters measured for determining fitness are listed twice in the abstract, I suggest that they only need listing once. 
  • AUTHORS: Thank you. The text was changed accordingly.

  • Some words are capitalised when they are not proper nouns. This is true throughout the paper, particularly when the tests are being listed.

  • AUTHORS: The text was checked and changed accordingly.

  • The consequence and meaning of the correlations is not interpreted in the abstract. A statement of what the results mean would be helpful as part of the conclusion of the abstract.

  • AUTHORS: Thank you much. The meaning of the correlation was added in the abstract. The text was checked and changed accordingly.

Introduction:

  • There is a mix of the word 'female' (sex) and 'woman/women' (gender). It may be helpful to clarify and ensure that the correct term is used.
  • AUTHORS: Done, finale we use female soccer players in all the text.

  • The sentence on lines 93-98 is very long and it is difficult to interpret what is being communicated. Please can this be reworked. Words such as 'it is necessary to mention' are usually superfluous and can be omitted without any lack of meaning from a sentence.
  • AUTHORS: The text was changed accordingly.

  • There seems to be a strong thread about the connection to injury and asymmetries that could lead to injury, and yet injury incidence or history is not collected.
  • AUTHORS: Thank you very much. In this case, we think that the present manuscript is a previous step to research the injury incidence. We only give a mention to the actual interest in this topic.

  • Line 121 - I think this should read 'male sex' and not 'male gender' Sex is biological, gender is expression of identify.

AUTHORS: Done. The text was changed accordingly.

Methods

  • Line 144 - what is meant by 'al analytic'?
  • AUTHORS: Sorry, we add the correct info. An observational analytic cohort design

  • Line 148 - what is 'sene'?
  • AUTHORS: The text was changed accordingly.

  • Line 150 - Should it read 'day' where it says 'daty'?
  • AUTHORS: Day The text was changed accordingly.

  • Line 151 – This should read 'All participants completed all sessions and assessments'. The word season is confused for session in several places in the paper.

AUTHORS:Thank you very much.  text was checked and changed accordingly.

  • Line 145 - the correct English would be 'were performed three times during the pre-season'.  and 'In the first season' should read 'In the first session'.
  • AUTHORS: Thank you very much. The text was changed accordingly.

  • Line 153 - two spelling/typographical errors.

  • AUTHORS: The text was changed accordingly.

  • Line 154 - The English in this sentence needs correcting, I'm not entirely clear what is meant here. I think it may mean 'During the period when assessments were undertaken, the participants were not playing any official matches.'

  • AUTHORS: The text was changed accordingly.

  • Line 155 - again the English translation is awkward and needs reworking. I think the keywords here are about technique (not 'correct grip').

  • AUTHORS: The text was changed accordingly.

  • Line 159-161 - spelling/typographical errors

  • AUTHORS: The text was changed accordingly.

  • Line 169 - spelling/typographical errors

  • AUTHORS: The text was changed accordingly.

  • Line 171 - spelling/typographical errors

  • AUTHORS: The text was changed accordingly.

  • Line 173 - 'an' should be 'and'

  • AUTHORS: The text was changed accordingly.

  • Line 176 - 'seasons' should be sessions, this is true further on in the paper and needs attending to. To save repeating this comment, please check all appearances of the word 'season' in the paper and check that it is correct. Many instances should be 'session'.

AUTHORS: Thank you very much. The document was checked and changed.

  • Line 181 - 'near of looker or with' needs correcting
  • AUTHORS: Thank you very much. The document was checked and changed.

  • Line 183 - the fitness status tests do not need listing again.

  • AUTHORS: Thank you very much. This paragraph was delete.

  • Throughout 2.3.1 - please clarify the type of temperature measurement (i.e., dry bulb) and the correct units for humidity need to be used (Line 187 'C' is used)

  • AUTHORS: The text was updated.

  • My view of methods is that it should be as clear as possible and not require significant reference to other papers. The citations in lines 181 and 191 do make it quite difficult to understand what is being carried out. With the large number of tests being used in this paper I would recommend that a figure/table/diagram is used to illustrate more clearly what is being measured and when. 

  • AUTHORS: Thank you very much. The diagram was added to elucidate the text.

  • Line 206 - 'Ipad' should read 'iPad'

  • AUTHORS: Done

  • The procedures are described (line 174-212) and then much is repeated when the measures are described (line 213-313). The repeat of information between the two sections could be improved for further conciseness and better structure.

  • AUTHORS: The manuscript was changed accordingly following your recommendations.

  • Line 232 – ‘Hop D’ and ‘Hop non-D’ need explaining and/or referencing.

  • AUTHORS: Done

  • Line 236 – “…previous suggestions by Rosch…” is too vague.

  • AUTHORS: Thank you very much. The text was updated accordingly.

  • Line 240 – The MySprint app needs full details.

  • AUTHORS: Thank you very much. The text was updated accordingly.

  • Line 243 - 'Ipad' should read 'iPad'

  • AUTHORS: Done

  • Line 244 – “To ensure successful performance, we followed the protocol of…” again, is too vague.

  • AUTHORS: The text was updated accordingly.

  • Line 261 – This is the first time ‘FMS’ is mentioned and therefore the full words need to be given alongside the abbreviation.

  • AUTHORS: The new information was added: Functional Movement Systems (FMS)

  • Line 266 – ‘…heel on the head.’? I’m not sure what this means.

  • AUTHORS: The text was updated accordingly.

  • Line 277 - the word 'participant' would be preferable to 'subject'

  • AUTHORS: The term was changed accordingly.

  • The Thomas Test is not clearly explained.

  • AUTHORS: The text was changed accordingly considering his feedback.

  • Line 311 – full details of the Kinovea Software is needed.

  • AUTHORS: The text was changed accordingly considering his feedback.

  • Line 315 - "The mean and standard deviation were used for data processing" it's unclear what this means and how the data was used. More detail is needed for clarity.

  • AUTHORS: The statistical procedures section was updated.

  • Table 1 - age should be presented to the same number of decimal places to that which it was originally collected - this is likely to be 0 decimal places. i.e., 21 ± 4. The same is true for the other measures, were they all recorded to 2 decimal places?

  • AUTHORS: Thank you very much. Normally, we presente the values of diferente tables with two decimals.

  • Line 341 - 'In this sense' needs amending, the phrasing is unclear.

  • AUTHORS: We use “in this respect” or “Accordingly” to understand the manuscript.

  • Table 2, 3, 4, 5, 6, 7 – I’m not sure what the tables are adding to the narrative of the paper. The inclusion of the individual data, without connecting to other data doesn’t help the reader understand the relationship between the fitness status and functional results. Reporting individual observational data doesn’t offer any insights, analysis or clear interpretation for discussion. Table 8 is a more useful table as it draws together the connections between the results. I would prefer to see the mean, SD, and ranges for the results, presented in a table – these would be useful observations and would add to the data about female soccer.

  • AUTHORS: Thank you very much. Finally, we use the table in Thomas test, hip extension (with knee extension and knee flexion) and Internal and external hip rotators assessment. The other values were shown in the text.

  • Discussion

  • Line 440-444 – It’s not necessary to repeat all the test names again.

  • AUTHORS: The text was changed accordingly.

  • Line 466-476 – the paragraph is written in a vague way that makes it difficult to understand and therefore it lacks a clarity of message. I’m not sure if the results of the study are being discussed in relation to the published research, or the solely the published research.

  • AUTHORS: The discussion was updated and changed following his suggest.  

  • Line 485 – this is the first time the women have been described as ‘semi-professional’, this should be mentioned earlier in the paper.

  • AUTHORS: Thank you very much. In the introdution section, we add information relevant to describe semi.professional female soccer players.

  • Line 498 – “the entire physical and physiological profile has been brought together in this research” – I don’t think this profile is well presented in the results. The data is available but not presented in such a way to give a clear profile.

  • AUTHORS: The text was changed accordingly. We present the means of diferents test and justify the values obtained and his importance.

  • Line 501 – a conclusion should be the profiling of these athletes, but only if the results are presented to give this more clearly.

  • AUTHORS: Thank you very much. The conclusion section was changed accordingly.

Reviewer 3 Report

Paper Title: A data analytics approach to assess the functional and physical 2

performance of female soccer players

Journal Manuscript ID: IJERPH - 1808276

REVIEW 

Thank you for the opportunity to review this manuscript.

I have read with interest and my impression is that it is a good job. The big problem is their sample size, which does not allow the results to be generalized. In addition, I suggest some changes to your paper. Here I show some of them. Good luck!

Title and Abstract

·       Title: Please, Indicate the study’s design

·       Abstract: please indicate background of the subject and precise conclusions. Add conclusions as to what the implications of your results are. i.e., What the implications of high asymmetries in some test of physical condition are? or What are the implications of correlations between these functional performance variables and the anthropometric and physical characteristics of athletes??

·       Try to give a balanced summary of what was done and what was found.

Introduction

·       Line 55, there is a bracket with two parentheses. This error is repeated throughout the text (lines 59, 50, 60, 62, 65, 68…etc.). Please remove the parentheses and leave only the references inside the brackets.

·       Include the target population in the objectives.

·       Add the hypothesis before aims. Also, include the target population in the hypothesis.

Methods

·       This section should begin by presenting key elements of study design early in the paper. In the study design section only provide information regarding the type of design, place of conduct and aspects related to ethical principles (such as ethics committee, informed consent, etc.).

·       Setting section: Describe the setting, locations, and relevant dates, including periods of recruitment, exposure, follow-up, and data collection.

·       Line 154: “momento” spelling mistake.

·       Participants section: please do not include analyzed data. This section is for methodology only. Data (number of participants, means, standard deviations, etc.) should be put in results. Also, the reference to table 1.

·       Include exclusion criteria in the participants' section

·       Line 213. I would include “Outcome Measures”

·       Line 246: the sentence appears in italics unlike the others of the same format.

·       Explain how the study size was arrived at.

Results

·       Please, start this section with the information that you had put in the participants section (the analyzed data).

Discussion

·       Please, do not forget to discuss limitations of the study, taking into account sources of potential bias or imprecision. Discuss both direction and magnitude of any potential bias.

Conclusions

·       In the conclusions section, please add what your findings mean or have implications for the world of sport and science.

References

·       The references do not follow the standards set by the magazine. Please check and change them.

Author Response

REVIEWER 3

Paper Title: A data analytics approach to assess the functional and physical 2

performance of female soccer players

Journal Manuscript ID: IJERPH - 180827

Thank you for the opportunity to review this manuscript.

I have read with interest and my impression is that it is a good job. The big problem is their sample size, which does not allow the results to be generalized. In addition, I suggest some changes to your paper. Here I show some of them. Good luck!

Title and Abstract

  • - Title: Please, Indicate the study’s design.

AUTHORS: Done

- Abstract: please indicate background of the subject and precise conclusions. Add conclusions as to what the implications of your results are. i.e., What the implications of high asymmetries in some test of physical condition are? or What are the implications of correlations between these functional performance variables and the anthropometric and physical characteristics of athletes?? Try to give a balanced summary of what was done and what was found.

AUTHORS:  The abstract were changed accordingly.

Introduction

-  Line 55, there is a bracket with two parentheses. This error is repeated throughout the text (lines 59, 50, 60, 62, 65, 68…etc.). Please remove the parentheses and leave only the references inside the brackets.

AUTHORS: Thank you very much.  The text was changed accordingly.

-  Include the target population in the objectives

AUTHORS: Done

- Add the hypothesis before aims. Also, include the target population in the hypothesis.

AUTHORS:  Done

Methods

- This section should begin by presenting key elements of study design early in the paper. In the study design section only provide information regarding the type of design, place of conduct and aspects related to ethical principles (such as ethics committee, informed consent, etc.).

AUTHORS: Thank you very much. The document was changed accordingly.

- Setting section: Describe the setting, locations, and relevant dates, including periods of recruitment, exposure, follow-up, and data collection.

AUTHORS:  Thank you very much for your feedback. The document was changed accordingly.

- Line 154: “momento” spelling mistake.

AUTHORS: Done

- Participants section: please do not include analyzed data. This section is for methodology only. Data (number of participants, means, standard deviations, etc.) should be put in results. Also, the reference to table 1.

AUTHORS: The document was changed accordingly.

-  Include exclusion criteria in the participants' section

AUTHORS: Done

- Line 213. I would include “Outcome Measures”

AUTHORS: The document was changed accordingly.

- Line 246: the sentence appears in italics unlike the others of the same format.

AUTHORS: Thank you vey much. The diferentes section were checked and all have the same format

-  Explain how the study size was arrived at.

AUTHORS: Thank you. In participants section the information was added.

Results

- Please, start this section with the information that you had put in the participants section (the analyzed data).

AUTHORS:  Done

Discussion

- Please, do not forget to discuss limitations of the study, taking into account sources of potential bias or imprecision. Discuss both direction and magnitude of any potential bias.

AUTHORS: Dear reviewer. The recomendations were taken in consideration and were added information in the final of discussion.

Conclusions

In the conclusions section, please add what your findings mean or have implications for.

AUTHORS: The conclusion section was updated following your recommendations.

References

  • The references do not follow the standards set by the magazine. Please check and change them.
  • AUTHORS: The references were updated and changed accordingly in all document.

Round 2

Reviewer 1 Report

Authors carried out revisions on the format and typing errors. However, the quality of this manuscript is still too low and presents serious flaws. It is just a descriptive analysis of a WSP. Authors use the interpretation values of effect sizes to interpret r values.... Authors can't differentiate between tests used to perform bivariate analyses and effect sizes. This paper can't be published in this journal

Author Response

REVIEWER 1

Authors carried out revisions on the format and typing errors. However, the quality of this manuscript is still too low and presents serious flaws. It is just a descriptive analysis of a WSP. Authors use the interpretation values of effect sizes to interpret r values.... Authors can't differentiate between tests used to perform bivariate analyses and effect sizes. This paper can't be published in this jornal

Thank you very much for all your comments. The comments are highly and positively received, and we appreciate a lot your effort in supporting us to get a better-quality paper giving us your useful comments and recommendations. 

The descriptive analysis is necessary to understand the characteristic of this population group. We show different analysis also to improve the knowledge about this important sample.

In respect of your comment about the English language, we have reviewed the full article and we have used a professional in English writing and spelling.

Reviewer 2 Report

Thank you for attending to my comments and suggestions. The manuscript is further improved. However, not all of the corrections that are noted as 'done' have been made. While most have, there are still some language issues. Examples include line 157, line 190, line 424, and instances where words are capitalised when they are not proper nouns. I have not listed all instances and the paper does need checking carefully.

The diagram on line 211 doesn't really offer a detailed schematic of the sequence of testing. The week in the calendar when tests were carried out is not really necessary, it is the sequencing of specific tests within a testing session that needs clarifying.

I have also noticed that 'weight' is often used where it should be 'mass'. Please could this also be corrected.

Thank you.

Author Response

Thank you for attending to my comments and suggestions. The manuscript is further improved. However, not all of the corrections that are noted as 'done' have been made. While most have, there are still some language issues. Examples include line 157, line 190, line 424, and instances where words are capitalised when they are not proper nouns. I have not listed all instances and the paper does need checking carefully.

Thanks, the comments are highly appreciated. In fact, we believe vehemently that the present research is a significant contribution to the existing literature on the subject. We would also like to apologize for the mistakes made regarding the English language. We can ensure that the English writing and spelling have been properly reviewed by professionals (please see the attached certificate).

The diagram on line 211 doesn't really offer a detailed schematic of the sequence of testing. The week in the calendar when tests were carried out is not really necessary, it is the sequencing of specific tests within a testing session that needs clarifying. Thank you. The diagram was changed accordingly.

I have also noticed that 'weight' is often used where it should be 'mass'. Please could this also be corrected. Thank you so much for your suggestion. The term has been changed accordingly.

Thank you.